# Declining but Pronounced State-Level Disparities in Prescription Opioid Distribution in the United States

**DOI:** 10.3390/pharmacy12010014

**Published:** 2024-01-16

**Authors:** Joshua D. Madera, Amanda E. Ruffino, Adriana Feliz, Kenneth L. McCall, Corey S. Davis, Brian J. Piper

**Affiliations:** 1Department of Medical Education, Geisinger Commonwealth School of Medicine, Scranton, PA 18509, USA; jmadera@som.geisinger.edu (J.D.M.); aruffino@som.geisinger.edu (A.E.R.); afeliz@som.geisinger.edu (A.F.); bjpiper1@geisinger.edu (B.J.P.); 2Department of Pharmacy Practice, University of New England, Portland, ME 04103, USA; 3Department of Pharmacy Practice, Binghamton University, Johnson City, NY 13790, USA; 4Network for Public Health Law, Edina, MN 55435, USA; cdavis@networkforphl.org; 5Center for Pharmacy Innovation and Outcomes, Geisinger College of Health Sciences, Danville, PA 18704, USA

**Keywords:** opiate, addiction, pain, pharmacoepidemiology, public policy

## Abstract

The United States (US) opioid epidemic is a persistent and pervasive public health emergency that claims the lives of over 80,000 Americans per year as of 2021. There have been sustained efforts to reverse this crisis over the past decade, including a number of measures designed to decrease the use of prescription opioids for the treatment of pain. This study analyzed the changes in federal production quotas for prescription opioids and the distribution of prescription opioids for pain and identified state-level differences between 2010 and 2019. Data (in grams) on opioid production quotas and distribution (from manufacturer to hospitals, retail pharmacies, practitioners, and teaching institutions) of 10 prescription opioids (codeine, fentanyl, hydrocodone, hydromorphone, meperidine, methadone, morphine, oxycodone, oxymorphone, and tapentadol) for 2010 to 2019 were obtained from the US Drug Enforcement Administration. Amounts of each opioid were converted from grams to morphine milligram equivalent (MME), and the per capita distribution by state was calculated using population estimates. Total opioid production quotas increased substantially from 2010 to 2013 before decreasing by 41.5% from 2013 (87.6 MME metric tons) to 2019 (51.3). The peak year for distribution of all 10 prescription opioids was between 2010 and 2013, except for codeine (2015). The largest quantities of opioid distribution were observed in Tennessee (520.70 MME per person) and Delaware (251.45) in 2011 and 2019. There was a 52.0% overall decrease in opioid distribution per capita from 2010 to 2019, with the largest decrease in Florida (−61.6%) and the smallest in Texas (−18.6%). Southern states had the highest per capita distribution for eight of the ten opioids in 2019. The highest to lowest state ratio of total opioid distribution, corrected for population, decreased from 5.25 in 2011 to 2.78 in 2019. The mean 95th/5th ratio was relatively consistent in 2011 (4.78 ± 0.70) relative to 2019 (5.64 ± 0.98). This study found a sustained decline in the distribution of ten prescription opioids during the last five years. Distribution was non-homogeneous at the state level. Analysis of state-level differences revealed a fivefold difference in the 95th:5th percentile ratio between states, which has remained unchanged over the past decade. Production quotas did not correspond with the distribution, particularly in the 2010–2016 period. Future research, focused on identifying factors contributing to the observed regional variability in opioid distribution, could prove valuable to understanding and potentially remediating the pronounced disparities in prescription opioid-related harms in the US.

## 1. Introduction

Opioid drug overdose deaths, largely driven by illicit fentanyl, recently exceeded 80,000 in one year, the highest number ever recorded in the US [1]. The US is disproportionately affected by the opioid epidemic, with an opioid-related mortality ten times higher than in some European countries [2]. Prescription opioids continue to have a strong presence throughout the US. The opioid prescription rate per capita increased by 7.3% between 2007 and 2012, with larger elevations in pain medicine (8.5%) and physical medicine/rehabilitation (12.0%) [3]. This profile has since been reversed. Following an overall peak in 2011, there has been a decline in opioid distribution in the US [4,5]. Despite these reductions, the amount of opioids prescribed remains elevated, and in 2015 [6] was almost three times higher than in 1999 [7]. The morphine milligram equivalent (MME) per capita in 2016 in the US was 1124 [4], which is appreciably higher than in other developed countries [8] or the US Territories [9]. Interstate variability in opioid distribution rates was also observed in 2016. There was a fivefold difference between the lowest (North Dakota = 485) and highest (Rhode Island = 2624) states [4]. Examination of individual opioids [10] revealed a threefold state-level difference for fentanyl [11] and an 18-fold difference for meperidine [12].

The primary aim of this study was to examine the variation in production quotas for and state-level distribution of 10 prescription opioids used primarily for pain between 2010 and 2019, which to our knowledge has not been reported in the literature. This range was chosen because it includes the implementation of legislation that mandated limits on the prescribing or dispensing of opioids for acute pain in 26 states [11,13]. The timeframe of the study was also selected because it is prior to the COVID-19 pandemic and associated drug shortages [14]. Identifying the states with the greatest and least declines in opioid distribution as well as quantifying this variation may be useful to inform public health research into the correlates and causes of opioid-related harm, as well as efforts to determine the effectiveness of efforts to reduce the impact of that harm.

## 2. Materials and Methods

### 2.1. Procedures

Opioid production quota data were obtained from the US Drug Enforcement Administration’s (DEA) Production Quotas System [15], and opioid distribution data from their Automation of Reports and Consolidated Orders System (ARCOS). ARCOS is a publicly available reporting system that reports detailed and comprehensive drug information from manufacturing to distribution (including, at the dispensing/retail level, hospitals, retail pharmacies, practitioners, mid-level practitioners, and teaching institutions) [16]. ARCOS data corresponds highly with the dispensing of opioids as reported to state prescription drug monitoring programs [4,17] and has been used in many prior investigations [4,5,9,10,11,12,13,14,15,17,18]. Ten Schedule II opioid prescription drugs commonly used for the treatment of pain (codeine, fentanyl base, hydrocodone, hydromorphone, meperidine, methadone, morphine, oxycodone, oxymorphone, and tapentadol) were collected. Buprenorphine [19] was excluded because it is primarily employed for the treatment of opioid use disorder (OUD). Methadone, when used for OUD from Narcotic Treatment Programs, was also excluded [10]. All procedures were approved by Geisinger and the University of New England’s Institutional Review Board. 

### 2.2. Data Analysis

The weight in grams of each opioid was obtained from ARCOS from 2010 to 2019 to isolate the peak opioid distribution year. For the analysis of changes over time, the a priori comparison was between the peak year for opioid distribution (2011) and 2019. The oral MME for each opioid drug was determined using the appropriate multipliers (codeine 0.15, fentanyl base 75, hydrocodone 1, hydromorphone 4, meperidine 0.1, methadone 8, morphine 1, oxycodone 1.5, oxymorphone 3, and tapentadol 0.4 [4,5]) and summed for all 10 opioids across all 50 states. Heat maps demonstrating the percent change across all 50 states per capita in total opioid distribution from 2011 to 2019 were produced using JMP V.16.2.0 software. The per capita distribution of each opioid drug for each state was calculated using the respective population estimate from the annual American Community Survey [20]. Ratios for the highest to lowest states and for the 95th:5th and 70th:30th percentiles were calculated using these values [21]. To calculate these ratios, the per capita distribution of each opioid drug was organized from largest to smallest values, and the 95th and 5th percentile values were found. Using these values, the ratios were calculated. The 95th:5th percentile ratio identifies the degree of variation in distribution that each opioid experienced within a particular year across the US, where a larger ratio signifies a greater extent of variation. Spearman correlations were calculated for each opioid pair. The standard error of the mean (SEM) depicted variability. States outside of a 95% confidence interval were identified. A Fisher r to Z transformation was completed to determine if the correlations differed between 2011 and 2019.

## 3. Results

### 3.1. Production

An examination of total opioid production quota data from 2010 (54.9 MME metric tons) to 2019 revealed a gradual increase from 2011 to 2012, followed by a steep increase from 2012 to 2013 (87.6 tons), which remained relatively steady until a pronounced decline from 2016 (85.2 tons) to 2017 (57.6 tons). There was a 39.8% reduction from 2016 to 2019 (51.3 tons). Meperidine (−60.0%), oxymorphone (−54.3%), morphine (−49.6%), fentanyl (−48.5%) and hydrocodone (−48.1%), exhibited pronounced declines. Oxycodone (−38.5%), codeine (−37.5%), hydromorphone (−32.9%), methadone (−30.1%), and tapentadol (−27.5%) had sizable but more modest reductions in opioid production quotas (Figure 1A).

### 3.2. Distribution 

#### 3.2.1. National Opioid Distribution by Drug

Analysis of national prescription opioid distribution from 2010 to 2019 identified 2011 as the peak year overall (Figure 1B), leading to our analysis on the differences between the peak year (2011) and 2019. The peaks for each drug were 2010 for methadone, oxycodone, and meperidine; 2011 for hydrocodone and oxymorphone; 2012 for morphine and tapentadol; 2013 for hydromorphone and fentanyl; and 2015 for codeine. Since 2011 (23.2 MME metric tons), the total opioid distribution for the 10 opioids for pain has declined. From 2011 to 2019 (12.6 tons), a total decline in the opioid distribution of 45.4% was observed, with 2018 to 2019 showing an 11.3% decline. Analysis of individual opioid distribution trends revealed codeine to have the largest per capita increase (+143.09%) and oxycodone to have the largest decrease (−50.09%) from 2011 to 2019.

#### 3.2.2. State-Level per Capita Total Opioid Distribution

Comparison of total opioid distribution per capita in 2011 and 2019 revealed an overall decrease (−51.96%), but also considerable state-level differences, illustrated in Figure 2A and Figure 3. 

In 2011, four states, Tennessee (520.70), Nevada (491.38), Delaware (476.15), and Florida (452.12), had significantly elevated levels of total opioid distribution relative to the national mean (284.34, *p* < 0.05, Figure 2A). However, only two states, Alabama (251.45) and Delaware (238.71), were significantly above the national average (162.09) in 2019. Forty-three states had an MME per capita > 200 in 2011 relative to only six in 2019 (χ2(1) = 54.78, *p* < 0.0001). Four of these six states were in the southeastern (KT, TN) or south-central (AL, OK) US. Furthermore, there was a 3.31-fold difference between the highest (Florida = −61.61%) and lowest (Texas = −18.64%) states for the decline in opioid distribution. Figure 2B demonstrates the observed percent difference for total opioid distribution during this period and shows that other states with >50% reductions were Nevada (−59.7%), Ohio (−57.4%), Tennessee (−55.2%), Maine (−54.4%), Oregon (−54.3%), and West Virginia (−50.9%).

#### 3.2.3. State-Level per Capita Variability in Opioid Distribution by Drug

The data for 2011, depicted in Figure 3A,C, were used to calculate the ratio between the maximum and the minimum state per capita MME for each of the individual opioids. 

The ratio of per capita MME distribution of opioids from the maximum to the minimum states was highest for oxymorphone (27.4), followed by meperidine (19.6), tapentadol (14.0), oxycodone (13.6), methadone (11.2), hydrocodone (9.2), hydromorphone (4.6), morphine (4.2), codeine (4.0), and fentanyl (2.5, mean = 11.0 ± 2.4). The data for 2019, depicted in Figure 3B,D, were used to calculate the ratio between the maximum and minimum state per capita MME for each of the individual opioids. This data again demonstrated highest for oxymorphone (38.9), but instead was followed by methadone (30.3), meperidine (22.2), hydrocodone (13.4), tapentadol (12.3), codeine (9.8), oxycodone (6.1), hydromorphone (4.9), morphine (3.2), and again lowest for fentanyl (3.1) (mean = 14.4 ± 3.7, t(9) = 1.45, *p* = 0.18).

Figure 4 shows the ratios between the 95th:5th percentiles for the population-corrected distribution of each Schedule II opioid for 2011 and 2019. The 2011 ratio was highest for oxymorphone (8.0) and meperidine (7.3) and smallest for codeine (2.4) and fentanyl (1.8). The 2019 ratio was highest for methadone (10.2) and oxymorphone (10.1) and lowest for morphine (2.21) and fentanyl (2.11) in 2019. A comparison of the ratios between 2011 (4.78 ± 0.70) and 2019 (5.64 ± 0.98) revealed a nonsignificant difference.

#### 3.2.4. Correlations in the per Capita MME Distribution between Each Opioid

Correlations in the MME per capita per state between each opioid are shown in Table 1. Oxycodone had the most significant correlations with the other opioids in both 2011 and 2019. 

In 2011, there were three, mostly moderate (r = 0.35–0.69), correlations between oxycodone and morphine, oxymorphone, and methadone. There were also moderate associations between hydrocodone and meperidine and between oxymorphone and tapentadol. In 2019, there were three moderate correlations between oxycodone with morphine, oxymorphone, and methadone. There were also moderate correlations between hydrocodone and meperidine as well as hydrocodone and codeine. Fisher r to Z statistics were calculated to identify differences in the strength of the correlations between opioids in 2011 relative to 2019. These analyses revealed a significant difference, i.e., an increase, in the magnitude of association between codeine and hydrocodone (*p* < 0.05), codeine and hydromorphone (*p* < 0.01), hydrocodone and methadone (*p* < 0.05), and methadone and oxycodone (*p* < 0.05). Comparisons between the averages of the correlations were then completed for each opioid with each other opioid. A significant difference was identified in the mean correlation between 2011 and 2019 for methadone (*p* < 0.05).

## 4. Discussion

### 4.1. Production

There are several novel findings from this investigation into the dynamic changes in the production quotas and distribution of prescription opioids in the US. Although there has been much prior attention to opioid distribution [3,4,5,6,7,9,10,11,12,13,14], the changes in the production quotas have received limited quantitative attention. The total production quotas were about three times higher than opioid distribution throughout the study period, suggesting that the quotas had little or no impact on the distribution of those medications and were not based on a systematic evaluation of medical need. One caveat is that distribution included methadone only when used for pain, while the production quota included methadone that was subsequently employed for both pain and narcotic treatment programs. By MME, methadone is by far the predominant prescription opioid in the US [4]. However, quotas for other opioid drugs, such as hydrocodone, oxycodone, and oxymorphone, also increased from 2013 to 2016, even though the distribution of those drugs decreased every year during that period, demonstrating a large disconnect between the production quotas and actual distribution. 

The total opioid production quota from 2010 to 2019 demonstrated a pronounced increase from 2013 to 2016. There was a 40% decline after 2016. This pattern contrasts with the steady decline in opioid distribution observed from 2010–2019. The difference in pattern of decline in opioid production and distribution may be due to factors such as the failure of DEA to ensure that quotas for production matched the medical need for the drugs (distribution). The DEA is required to set these aggregate production quotas (APQs), one of three types of quotas set by the Administration, each year at levels that provide for the “estimated medical, scientific, research and industrial needs of the United States” and related purposes [22]. As the Department of Justice’s Office of the Inspector General described in a 2019 report, the DEA failed to ensure that the quotas for many opioids were consistent with actual medical need for the drugs, and in 2017 administratively reduced the APQs for most controlled substances by 25% [23]. This change to the APQ could contribute to the opioid production decline observed in 2017, which is now similar to the observed decline in opioid distribution. The DEA modified its regulations in August 2018 to change the factors for the DEA to consider in making determinations about the quota, including “the extent of any diversion of the controlled substance in the class” as well as relevant information from the FDA, CDC, CMS, and the states [24]. Enhanced collaboration between the DEA and agencies in the Department of Health and Human Services may improve the systematic evaluation of medical needs when setting appropriate quotas.

### 4.2. Distribution

This study determined that the distribution of prescription opioids for pain has undergone a pronounced reduction over the past decade, which is congruent with and extends upon reports from earlier periods [3,4,5,6,7,11]. This report also identified a consistent decline in opioid distribution since 2011 [4]. A myriad of reasons may explain this decline, including increased awareness of the addictive nature of opioids [25], continued escalation in opioid-related overdose mortalities [1], and the increased government funding and resources resulting from the classification of the opioid crisis as a public health emergency [26]. Evidence has been more subtle or contradictory for any measurable impact of prescription drug monitoring programs [27,28] or state opioid prescribing laws [12,15,29].

Relative to the other nine opioids, oxycodone distribution was the most pronounced and also demonstrated the largest decline in opioid distribution relative to its peak year. This appreciable decline could be due to factors such as pharmaceutical market changes, as exemplified in the 2010 introduction of abuse-deterrent OxyContin. Two years after the 2010 formulation change, the dispensing rate for extended-release oxycodone decreased by 39% [30]. 

Although there have been overall reductions in opioid distribution, each opioid experienced varying degrees of distribution in a given year. The high-potency opioid fentanyl demonstrated the lowest variation (2011 percentile ratio = 1.77, 2019 ratio = 2.12), which extends upon prior reports [5,11]. Methadone rose to have the highest degree of variation in 2019, despite being third highest in 2011. The observed variation could be due to the lack of federal quantity limits on Schedule II drug prescriptions, which allow individual states and health insurance carriers to impose their own limitations regarding the prescribing and dispensing of Schedule II drugs. These restrictions on drug dispensation vary greatly due to limitations on drug days’ supply, dosage unit, and whether the restrictions apply only to the first prescription [31,32]. Furthermore, formulation changes, such as the one seen with oxycodone [30], could play a role in the observed differences in the degree of variation in distribution for a particular opioid in a particular year. However, the federal reclassification of hydrocodone from a Schedule III to a Schedule II drug in 2014 [33] did not appreciably impact the variation (2011 percentile ratio = 6.11, 2019 ratio = 7.99).

Similarly, opioid distribution was not homogeneous, which is consistent with prior studies [4,5]. Total per capita opioid distribution from 2011 was significantly elevated in Tennessee, Nevada, Delaware, and Florida relative to the national mean. When analyzing the total per capita opioid distribution in 2019, only two states, Alabama and Delaware, were significantly higher than the national average. Further, when analyzing the data nationally, Florida had the highest per capita opioid distribution decline, while Texas had the lowest decline [34]. In the case of Florida, the decline in opioid prescribing during the timeframe of our study may in part be due to the implementation of state legislation limiting the days’ supply of opioid prescriptions for acute pain [35]. We can speculate that the general allowance of state discretion on what policies to implement and how to implement/enforce them may play a role in these observed differences in progress toward decreasing opioid distribution. 

There were several positive correlations between opioid distribution per person per state. Interestingly, both 2011 and 2019 showed multiple associations with oxycodone. Strong correlations were also found for codeine and hydrocodone, codeine and hydromorphone, hydrocodone and methadone, and methadone and oxycodone. It is very likely that medical specialty preferences exist for the choice of opioid medication based on knowledge of opioid pharmacology, comfort level, and experience. Additional studies should explore the influence on treatment decisions at the patient level to further explore these opioid associations.

### 4.3. Limitations and Conclusions

While we identified changes in Schedule II opioid production quotas and distribution between 2011 and 2019 and greatly extended upon prior investigation [3,4,9,10,11,12,13,23,36] to explore the substantial state and regional variation in opioid distribution [37], this study was not without caveats. We reported a decline in opioid distribution beginning in 2011, based on the distribution rates of the opioids, as mentioned earlier. However, another commonly prescribed opioid, buprenorphine [19], was not included in this analysis and may have influenced to a small degree the peak year of opioid distribution. Another limitation is that ARCOS does not differentiate between dosage formulations in the quantity of each opioid reported in the database. Fentanyl is the most potent prescription opioid included in our analysis; however, the US CDC does not currently distinguish between pharmaceutical and illicit fentanyl in its reporting of opioid overdose deaths [1]. In addition, ARCOS data do not account for Schedule IV or V opioids that could influence the total opioid distribution across the US [38]. For example, tramadol was moved from a non-controlled medication to Schedule IV in 2014 [39]. The MMEs per capita for opioids primarily employed for pain would be even higher with the inclusion of additional non-Schedule II opioids. Furthermore, the per capita analysis was calculated using the total population and for each state rather than a subset of the population using prescription opioids. The proportion of persons in each state who use opioids may differ, which may limit the generalizability of our analysis. Moreover, the effects of racial, gender, and socioeconomic disparities were not examined here, but have been explored by others [40,41,42,43,44]. Stratifying data based on these demographics could identify more specific changes and further characterize continued disparities in opioid distribution. Importantly, this study extends upon prior ARCOS reports [4,5,9,10,11,12,13,14] and examines total opioid distribution (e.g., pharmacy, hospital, provider, and mid-level provider), although this analysis cannot distinguish between appropriate and inappropriate opioid use. Although opioid use and diversion are well recognized as occurring in hospitals [45,46,47], a future investigation could examine state-level disparities for pharmacy-distributed opioids. 

In conclusion, this study found dynamic changes in prescription opioid production quotas and sizable but gradual reductions in distribution across the US from 2010 to 2019. The production quotas seem to have little or no correspondence with medical needs for opioid medications, as expressed in the amount of opioids distributed. Further, the non-homogenous nature of opioid distribution and its effects on the population pose a barrier to continued efforts to one day overcome the US opioid epidemic. Further investigations of state- and county-level differences may inform the policies of opioid stewardship programs.

## Figures and Tables

**Figure 1 pharmacy-12-00014-f001:**
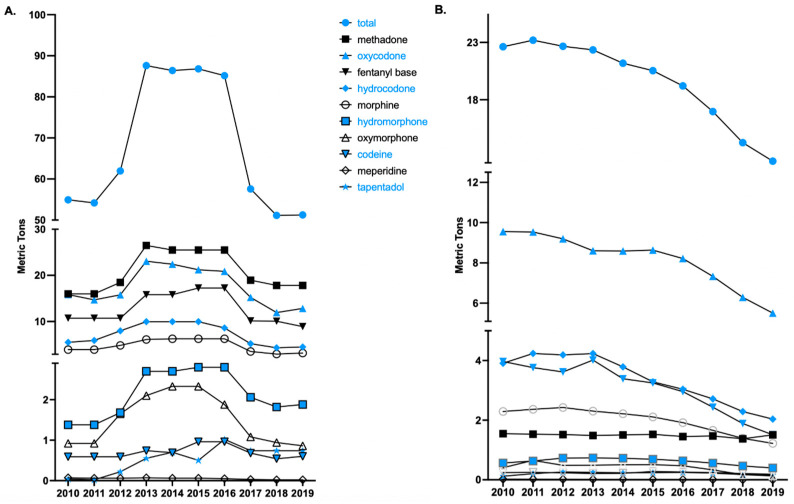
Total opioid production quotas (**A**) and distribution (**B**) in morphine mg equivalents (MME), as reported to the Drug Enforcement Administration across the United States from 2010 to 2019, of 10 Schedule II opioids used primarily for pain.

**Figure 2 pharmacy-12-00014-f002:**
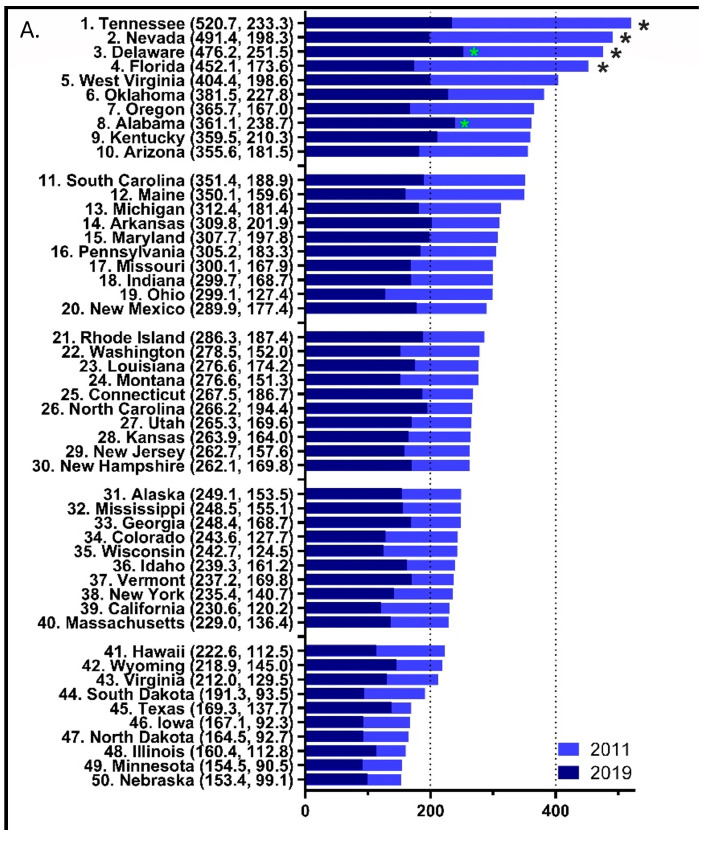
Per person distribution as reported to the US Drug Enforcement Administration’s Automated Reports and Consolidated Orders System was organized from largest to smallest values for 2011 with the 2019 per person distribution superimposed (**A**) with the values in 2019 and 2011 in parentheses after each state (* *p* < 0.05 versus the average states in 2011 (black font) and 2019 (green font)). Heat map percent reduction in total opioid distribution from 2011 to 2019 (**B**).

**Figure 3 pharmacy-12-00014-f003:**
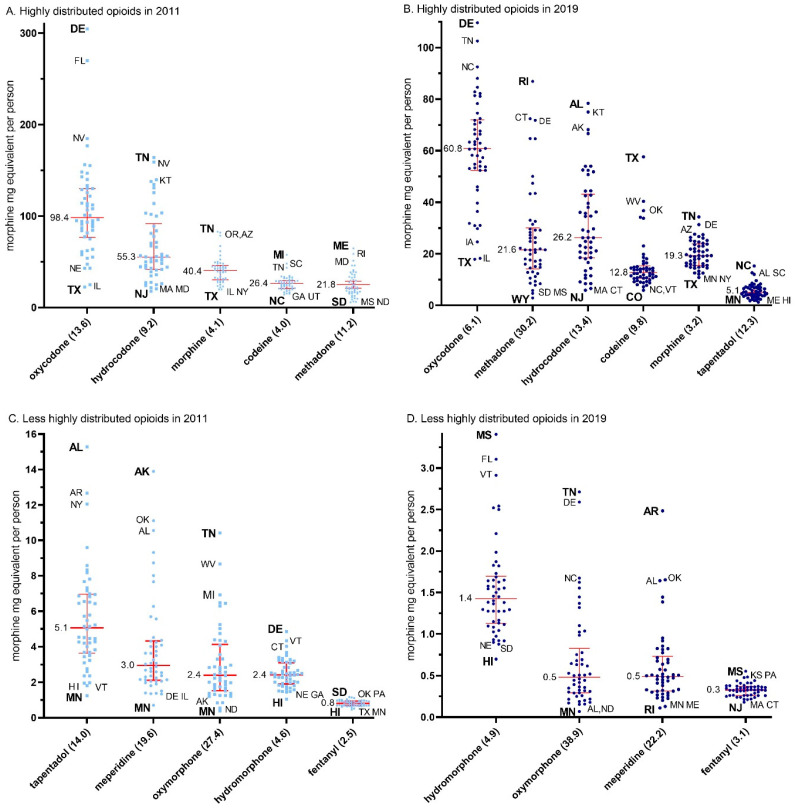
Median (+/− interquartile range) per person distribution as reported to the US Drug Enforcement Administration’s Automated Reports and Consolidated Orders System for highly (**A**,**B**) and less highly (**C**,**D**) distribution prescription opioids by morphine mg equivalents per state in 2011 (**A**,**C**) and 2019 (**B**,**D**). The maximum to minimum ratio is listed in parentheses after each opioid. The median is written next to each (red) solid horizontal line. Abbreviations are shown for states outside of the 95% confidence interval, with the highest and lowest states in bold.

**Figure 4 pharmacy-12-00014-f004:**
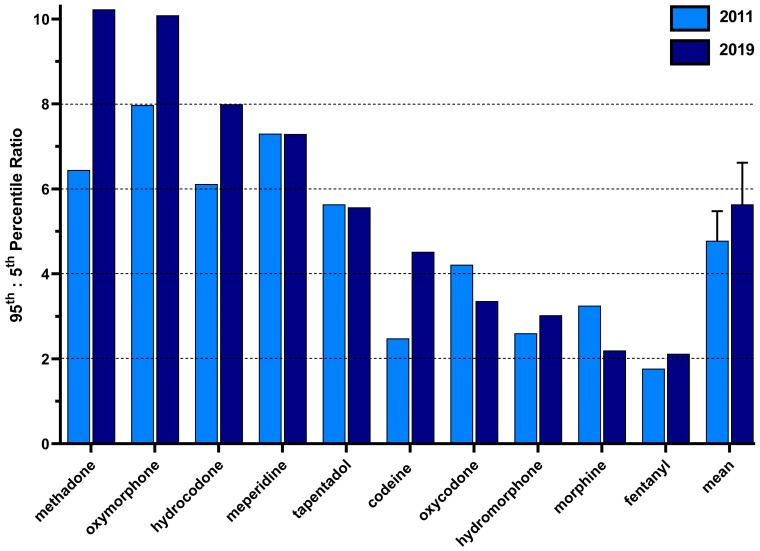
The 95th to 5th percentile ratio of the per person distribution as reported to the US Drug Enforcement Administration’s Automated Reports and Consolidated Orders System per state for each prescription opioid in 2011 and 2019.

**Table 1 pharmacy-12-00014-t001:** Correlations for per capita state opioid distribution as reported to the US Drug Enforcement Administration’s Automated Reports and Consolidated Orders System in 2011 and 2019. Boxes in blue and orange represent significant correlations between opioids of <0.05 and <0.01, respectively. * Significance between Z scores when comparing 2011 and 2019 correlations. ** Significance between mean correlations between 2011 and 2019.

**2011**	A	B	C	D	E	F	G	H	I	Mean	Standard Error
codeine (A)	1.00									0.15	0.05
fentanyl (B)	0.31	1.00								0.23	0.04
hydrocodone (C)	0.17 *	0.16	1.00							0.13	0.09
hydromorphone (D)	0.16 *	0.28	−0.30	1.00						0.13	0.09
meperidine (E)	−0.10	0.10	0.62	−0.22	1.00					0.10	0.09
methadone (F)	0.08	0.06	−0.15 *	0.40	−0.20	1.00				0.16	0.10
morphine (G)	0.33	0.39	0.24	0.29	0.18	0.35	1.00			0.26	0.07
oxycodone (H)	0.19	0.26	−0.12	0.44	0.02	0.66 *	0.50	1.00		0.29	0.09
oxymorphone (I)	0.25	0.36	0.32	0.18	0.16	0.35	0.28	0.50	1.00	0.32	0.04
tapentadol (J)	0.01	0.14	0.24	−0.04	0.38	−0.08	−0.21	0.13	0.51	0.12	0.08
**2019**	A	B	C	D	E	F	G	H	I	Mean	Standard Error
codeine (A)	1.00									0.08	0.08
fentanyl (B)	0.07	1.00								0.15	0.06
hydrocodone (C)	0.54 *	0.33	1.00							0.12	0.12
hydromorphone (D)	−0.31 *	0.14	−0.33	1.00						0.06	0.08
meperidine (E)	0.17	0.29	0.58	−0.07	1.00					0.17	0.09
methadone (F)	−0.10	−0.26	−0.45 *	0.31	−0.39	1.00				−0.02 **	0.10
morphine (G)	0.13	0.31	0.26	0.30	0.26	0.13	1.00			0.25	0.05
oxycodone (H)	0.10	0.05	0.02	0.25	0.21	0.39 *	0.51	1.00		0.29	0.07
oxymorphone (I)	0.12	0.24	0.14	0.16	0.10	0.23	0.29	0.61	1.00	0.27	0.06
tapentadol (J)	−0.03	0.21	0.03	0.05	0.34	−0.02	0.02	0.43	0.49	0.17	0.07

## Data Availability

All raw data are publicly available at: https://www.deadiversion.usdoj.gov/arcos/retail_drug_summary/index.html (accessed on 1 May 2023).

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
