# Peer review of "Declining but Pronounced State-Level Disparities in Prescription Opioid Distribution in the United States"

_pharmacy, 2024, doi:10.3390/pharmacy12010014_

Round 1
Reviewer 1 Report
Comments and Suggestions for Authors
Thank you for submitting the manuscript. I read with great interest your manuscript which deals with a really hot topic. Opioids are powerful and very useful drugs, unfortunately sometimes demonized for no reason. Your manuscript is interesting but in my opinion some changes are needed to make it more readable.
1)First of all, I ask you to use the MDPI format for the papers: this makes them readable and easily accessible for reviewers.
2) Your introduction does not focus enough on the fact that the opioid epidemic is a phenomenon that does not affect all countries. In fact, in some countries the situation is completely the opposite. I therefore ask you to read and use the following references: doi: 10.1186/s13052-021-00967-z. doi: 10.3390/ijerph191811754. doi: 10.1016/S2468-2667(19)30156-2.
3)The quality of your images is unacceptable. I ask you to make them better as they are blurry, faded and illegible.
I hope my comments are useful to you.
Kind Regards
Author Response
January 4, 2024
RE: Manuscript ID: pharmacy-2722036
Gary Zhang
Managing Editor, MDPI
E-mail: gary.zhang@mdpi.com
Dear Mr. Zhang,
We are delighted to resubmit our manuscript “Declining but Pronounced State Level Disparities in Prescription Opioid Distribution in the United States” for consideration by Pharmacy. We appreciate the thorough and insightful reviewer comments. Major revisions were made to address each reviewer’s comment and improve the clarity of the paper. Each reviewer’s comment is addressed below. Revisions and additions are tracked in the manuscript.
Thank you for your consideration of this important work.
Respectfully,
Kenneth L. McCall, PharmD, FAPhA
Clinical Professor & Chair
Binghamton University
School of Pharmacy and Pharmaceutical Sciences
96 Corliss Ave
Johnson City, NY 13790
kmccall@binghamton.edu
Reviewer #1
Thank you for submitting the manuscript. I read with great interest your manuscript which deals with a really hot topic. Opioids are powerful and very useful drugs, unfortunately sometimes demonized for no reason. Your manuscript is interesting but in my opinion some changes are needed to make it more readable.
1)First of all, I ask you to use the MDPI format for the papers: this makes them readable and easily accessible for reviewers.
Response: Thank you for your comments. A clean copy of the revised manuscript has been formatted into the MDPI standard. References throughout the manuscript were standardized to brackets as indicated in the instructions for authors.
2) Your introduction does not focus enough on the fact that the opioid epidemic is a phenomenon that does not affect all countries. In fact, in some countries the situation is completely the opposite. I therefore ask you to read and use the following references: doi: 10.1186/s13052-021-00967-z. doi: 10.3390/ijerph191811754. doi: 10.1016/S2468-2667(19)30156-2.
Response: Thank you for suggesting these references. An additional citation was added, and the first paragraph of the introduction was modified as follows:
Drug overdoses, largely driven by opioids, recently exceeded eighty thousand in one-year, the highest number ever recorded in the US [1]. The US is disproportionately affected by the opioid epidemic with a ten-fold higher opioid-related mortality than some European countries [2].
- Seth P, Rudd RA, Noonan RK, Haegerich TM. Quantifying the Epidemic of Prescription Opioid Overdose Deaths. Am J Public Health. (2018) 108(4):500-502.
3)The quality of your images is unacceptable. I ask you to make them better as they are blurry, faded and illegible.
Response: Thank you for this concern. Improved resolution images have been uploaded.

Reviewer 2 Report
Comments and Suggestions for Authors
The manuscript aims to presents changes in production quotas and distribution of selected opioid drugs in the US. The manuscript focuses on the time period 2010-2019 corresponding to opioid crisis. The presented results show an increase and subsequent decline in distribution of these agents with distinct differences in various American states. The main strength of the manuscript are as follows: analyzing numerous opioid drugs, presenting data on different levels, i.e. differences in years and states, explaining variation between states. Moreover authors employed rigorous methodology, explained inclusion and exclusion criteria, used proper statistical methods to assess data and described the observed trends both qualitatively and quantitatively. Data presentation is clear and consistent, the article is well written, all the presented data is discussed. The conclusions are consistent with the presented evidence, and since the authors used official data, their results are reproducible. The authors cited much literature, most of which is recent, but since authors write about period since 2010, referring to some older articles is also understandable. The authors identified further knowledge gaps that could be addressed by further studies.
I would raise some points that could be better addressed by authors:
1) The authors addressed an important topic, i.e. the appropriateness of regulatory affairs in light of the development of an opioid crisis in the US and relation between the production quotas and observed distribution. The research gap is well stated, however the primary hypothesis of the study could be better defined.
2) In lines 178-185 authors identified certain states which differed from others in distribution pattern. This issue was further addressed in lines 282-289, but without giving a clear explanation of the observed differences. Could authors speculate on what did these American states have in common and what could be the reason for their distinct distribution patterns?
3) One of the study limitation, I believe, that was not mentioned in the lines 297-313, is the fact that the ACROS data does not include various dosage forms. Some of them, are more likely to contribute to overdose as compared to others. The danger of overdose also depends on the potency of selected APIs and their affinity to receptors. Could authors point out which they tracked are the most important in relation to opioid crisis from the pharmacoepidemiology point of view?
4) Authors point out failure of federal agencies to set appropriate quotas for opioids correlating with the medical needs (line 246 and further 259-262). Could authors write a suggestion on improving decision making in this regard in future, based on the result of their study? What factors should be considered in systemic evaluation of medical needs?
5) The authors cited a work describing opioid distribution in hospitals in period overlapping with the one in the study (reference 4) as well as a study of similar opioids in a slightly different time period (2006-2016, reference 3). Could authors discuss their findings in relation to the findings and conclusions of the cited study? Do they draw a different picture?
I would also point out some minor issues:
1) In line 139 of the manuscript, the abbreviation JMP is not explained.
2) In lines 132-133 a sentence starting with ‘The weight in grams’ is repeated in lines 137-138
Author Response
January 4, 2024
RE: Manuscript ID: pharmacy-2722036
Gary Zhang
Managing Editor, MDPI
E-mail: gary.zhang@mdpi.com
Dear Mr. Zhang,
We are delighted to resubmit our manuscript “Declining but Pronounced State Level Disparities in Prescription Opioid Distribution in the United States” for consideration by Pharmacy. We appreciate the thorough and insightful reviewer comments. Major revisions were made to address each reviewer’s comment and improve the clarity of the paper. Each reviewer’s comment is addressed below. Revisions and additions are tracked in the manuscript.
Thank you for your consideration of this important work.
Respectfully,
Kenneth L. McCall, PharmD, FAPhA
Clinical Professor & Chair
Binghamton University
School of Pharmacy and Pharmaceutical Sciences
96 Corliss Ave
Johnson City, NY 13790
kmccall@binghamton.edu
Reviewer #2
Comments and Suggestions for Authors
The manuscript aims to presents changes in production quotas and distribution of selected opioid drugs in the US. The manuscript focuses on the time period 2010-2019 corresponding to opioid crisis. The presented results show an increase and subsequent decline in distribution of these agents with distinct differences in various American states. The main strength of the manuscript are as follows: analyzing numerous opioid drugs, presenting data on different levels, i.e. differences in years and states, explaining variation between states. Moreover authors employed rigorous methodology, explained inclusion and exclusion criteria, used proper statistical methods to assess data and described the observed trends both qualitatively and quantitatively. Data presentation is clear and consistent, the article is well written, all the presented data is discussed. The conclusions are consistent with the presented evidence, and since the authors used official data, their results are reproducible. The authors cited much literature, most of which is recent, but since authors write about period since 2010, referring to some older articles is also understandable. The authors identified further knowledge gaps that could be addressed by further studies.
I would raise some points that could be better addressed by authors:
- The authors addressed an important topic, i.e. the appropriateness of regulatory affairs in light of the development of an opioid crisis in the US and relation between the production quotas and observed distribution. The research gap is well stated, however the primary hypothesis of the study could be better defined.
Response: The second paragraph of the introduction was modified as follows:
The primary aim of this study was to examine the variation in production quotas for and state-level distribution of ten prescription opioids used primarily for pain between 2010 and 2019 which to our knowledge has not been reported in the literature. This range was chosen as it includes the implementation of legislation that mandated limits on the prescribing or dispensing of opioids for acute pain in twenty-six states [12,14]. The timeframe of the study was also selected as it is prior to the COVID-19 pandemic and associated drug shortages [15].
- In lines 178-185 authors identified certain states which differed from others in distribution pattern. This issue was further addressed in lines 282-289, but without giving a clear explanation of the observed differences. Could authors speculate on what did these American states have in common and what could be the reason for their distinct distribution patterns?
Response: The following sentence and citation were added to the discussion section to offer a further explanation of the observed differences.
Further, when analyzing the data nationally, Florida had the highest per capita opioid distribution decline while Texas had the lowest decline [35]. In the case of Florida, the decline in opioid prescribing during the timeframe of our study may in part be due to the implementation of state legislation limiting the days-supply of opioid prescriptions for acute pain [36].
- Geller JS, Milner JE, Pandya S, et al. The impact of the Florida law HB21 on opioid prescribing patterns after spine surgery. N Am Spine Soc J. 2023;14:100202. Published 2023 Feb 17. doi:10.1016/j.xnsj.2023.100202
- One of the study limitation, I believe, that was not mentioned in the lines 297-313, is the fact that the ACROS data does not include various dosage forms. Some of them, are more likely to contribute to overdose as compared to others. The danger of overdose also depends on the potency of selected APIs and their affinity to receptors. Could authors point out which they tracked are the most important in relation to opioid crisis from the pharmacoepidemiology point of view?
Response: Thank you for recognizing this limitation. The following sentences were added to the discussion section of the manuscript.
Another limitation is that ARCOS does not differentiate between dosage formulations in the quantity of each opioid reported in the database. Fentanyl is the most potent prescription opioid included in our analysis; however, the US CDC does not currently distinguish between pharmaceutical and illicit fentanyl in their reporting of opioid overdose deaths [1].
- Authors point out failure of federal agencies to set appropriate quotas for opioids correlating with the medical needs (line 246 and further 259-262). Could authors write a suggestion on improving decision making in this regard in future, based on the result of their study? What factors should be considered in systemic evaluation of medical needs?
Response: The authors support the enhanced interagency collaboration between the DEA, CDC, FDA and CMS when setting opioid production quotas. The following section of the discussion was expanded.
The DEA modified its regulations in August 2018 to change the factors for the DEA to consider in making determinations about the quota, including “the extent of any diversion of the controlled substance in the class” as well as relevant information from FDA, CDC, CMS and the states [31]. Enhanced collaboration between the DEA and agencies in the Department of Health and Human Services may improve the systematic evaluation of medical needs when setting appropriate quotas.
5) The authors cited a work describing opioid distribution in hospitals in period overlapping with the one in the study (reference 4) as well as a study of similar opioids in a slightly different time period (2006-2016, reference 3). Could authors discuss their findings in relation to the findings and conclusions of the cited study? Do they draw a different picture?
Response: Both the overall reduction in opioid use and the variability in per capita opioid use were consistent with findings from prior studies. The following sentences in the discussion section were modified to reflect this.
Although there have been overall reductions in opioid distribution, each opioid experienced varying degrees of distribution in a given year. The high-potency opioid fentanyl demonstrated the lowest variation (2011 percentile ratio = 1.77, 2019 ratio = 2.12) which extends upon prior reports [5,11,12].
Similarly, opioid distribution was not homogeneous which is consistent with prior studies [4,5].
I would also point out some minor issues:
- In line 139 of the manuscript, the abbreviation JMP is not explained.
Response: A clarification was added regarding JMP statistical software.
- In lines 132-133 a sentence starting with ‘The weight in grams’ is repeated in lines 137-138
Response: The redundant sentence was deleted. Thank you.

Reviewer 3 Report
Comments and Suggestions for Authors
Please see attached.

The language in this manuscript is of appropriate quality.
Author Response
January 4, 2024
RE: Manuscript ID: pharmacy-2722036
Gary Zhang
Managing Editor, MDPI
E-mail: gary.zhang@mdpi.com
Dear Mr. Zhang,
We are delighted to resubmit our manuscript “Declining but Pronounced State Level Disparities in Prescription Opioid Distribution in the United States” for consideration by Pharmacy. We appreciate the thorough and insightful reviewer comments. Major revisions were made to address each reviewer’s comment and improve the clarity of the paper. Each reviewer’s comment is addressed below. Revisions and additions are tracked in the manuscript.
Thank you for your consideration of this important work.
Respectfully,
Kenneth L. McCall, PharmD, FAPhA
Clinical Professor & Chair
Binghamton University
School of Pharmacy and Pharmaceutical Sciences
96 Corliss Ave
Johnson City, NY 13790
kmccall@binghamton.edu
Reviewer 3
- OVERALL: Thank you for the opportunity to review this manuscript. I appreciate the topic area of the manuscript, particularly related to federal production quotas. To ensure that the manuscript has optimal impact, I would recommend that the authors work on revisions as I have suggested below. Specifically, I’d recommend streamlining the text so that the clinical need, study mechanisms, and context of the findings are clear for the reader. This includes ensuring that information is placed in the right subsections of the manuscript so the reader can follow the study logically throughout. It also includes thinking through the best ways to present the data with sufficient but not overwhelming detail. I get the sense that this manuscript has been retrofitted or reformatted from an alternate submission (although perhaps not), so I’d also ask the authors to take care in reading the document top to bottom to remove redundancies and non-sensical text that may have resulted.
Response: Thank you for this valuable feedback. We have carefully addressed each comment below and the edits have improved the manuscript.
- TITLE: The phrase of ‘declines but pronounced’ reads a bit oddly – I’m not sure if ‘declining but pronounced’ would be accurate?
Response: Thank you. The title was modified accordingly.
- ABSTRACT:
The methods might benefit from a bit further explanation, namely what is meant by ‘distribution’ (to where/who?) The sentence provided only addresses how the data was sourced, but additional information on how it was analyzed would be useful – namely how data was represented (metric tons, MME per person, etc.) and compared (ratios, % changes, etc.). Population data was utilized to calculate per person estimates, which should be included. Essentially, there should be a rebalancing of the abstract to reduce the discussion (not repeating results) and lengthen the methods (so that the reader can better understand what was done with the data).
Response: The methods section of the abstract was expanded as follows.
Methods: Data (in grams) on opioid production quotas and distribution (from manufacturer to hospitals, retail pharmacies, practitioners and teaching institutions) of ten prescription opioids (codeine, fentanyl, hydrocodone, hydromorphone, meperidine, methadone, morphine, oxycodone, oxymorphone, and tapentadol) for 2010 to 2019 were obtained from the US Drug Enforcement Administration. Amounts of each opioid were converted from grams to morphine-mg-equivalent (MME) and the per capita distribution by state was calculated using population estimates.
INTRODUCTION:
- I would make clear at the number of drug overdoses mentioned is a US-specific estimate. The first paragraph contains a lot of useful data, but it lacks a coherent flow that helps tell the story. The mention of drug overdoses (while useful) at the start should be supplemented with some level of explanation that these days, this is primarily due to IMF (not prescription opioids). I would remove mention of buprenorphine state-level differences from the end of the first paragraph given it is not part of the current analysis.
Response: The first paragraph of the introduction was modified as follows to improve flow and address the comments above.
Opioid drug overdose deaths, largely driven by illicit fentanyl, recently exceeded eighty thousand in one-year, the highest number ever recorded in the US [1]. The US is disproportionately affected by the opioid epidemic with a ten-fold higher opioid-related mortality than some European countries [2]. Prescription opioids continue to have a strong presence throughout the US. The opioid prescription per capita increased by 7.3% between 2007 and 2012 with larger elevations in pain medicine (8.5%) and physical medicine/rehabilitation (12.0%) [3]. This profile has since reversed. Following an overall peak in 2011, there has been a decline in opioid distribution in the US [4,5]. Despite these reductions, the amount of opioids prescribed remains elevated, and in 2015 [6] was almost three-fold higher than 1999 [7]. The morphine milligram equivalent (MME) per capita in 2016 in the US was 1,124 [4], which is appreciably higher than other developed countries [8] or the US Territories [9]. Inter-state variability in opioid distribution rates was also observed in 2016. There was a five-fold difference between the lowest (North Dakota = 485) and highest (Rhode Island = 2,624) states [4]. Examination of individual opioids [10,11] revealed a three-fold state-level difference for fentanyl [12] and eighteen-fold difference for meperidine [13].
- The authors mention “implementation of legislative changes in most states intended to limit the distribution of opioids.” – this sentence would benefit from a bit more context. What types of changes are being referenced? The citation is very helpful in explaining this, but a (x) of examples would help provide context to this introduction, and connect perhaps to some of the trends noted later on in the results.
It also is not clear what gap in the literature exists. The first paragraph reports declines in opioid distribution since 2011 and interstate variability. How does this data set up what is known and unknown about prescription opioid distribution (and the need for the current analysis)?
Much of the second paragraph (past the objective statement) is methods – explaining the time range, where the data was obtained, reference to supplemental tables, explanation of the reporting system. This should be moved and integrated within this section.
Overall, the introduction should give the reader a sense of what has already been investigated, the importance of learning more, and what gap this study aims to fill in. The authors also might consider some description of what ‘opioid quotas’ are given not all readers may be familiar.
Response: The second paragraph of the introduction was substantially modified as follows to further clarify the literature gap and move statements regarding methods to the next section.
The primary aim of this study was to examine the variation in production quotas for and state-level distribution of ten prescription opioids used primarily for pain between 2010 and 2019 which to our knowledge has not been reported in the literature. This range was chosen as it includes the implementation of legislation that mandated limits on the prescribing or dispensing of opioids for acute pain in twenty-six states [12,14]. The timeframe of the study was also selected as it is prior to the COVID-19 pandemic and associated drug shortages [15]. Identifying the states with the greatest and least declines in opioid distribution as well as quantifying this variation may be useful to inform public health research into the correlates and causes of opioid-related harm as well as efforts to determine the effectiveness of efforts to reduce the impact of that harm.
METHODS:
- The methods generally benefit from beginning with the study design and data source (see my comment above about moving out of the introduction).
Response: The first paragraph of the methods section was modified as follows:
Opioid production quota data were obtained from the US Drug Enforcement Administration’s (DEA) Production Quotas System [16] (Supplemental Table 1), and opioid distribution data from their Automation of Reports and Consolidated Orders System (ARCOS). ARCOS is a publicly available reporting system that reports detailed and comprehensive drug information from manufacturing to distribution [17]. ARCOS data corresponds highly with dispensing of opioids as reported to state prescription drug monitoring programs [4,18] and has been used in many prior investigations [4,5,9-16,18,19]. Ten Schedule II opioid prescription drugs commonly used for the treatment of pain: codeine, fentanyl base, hydrocodone, hydromorphone, meperidine, methadone, morphine, oxycodone, oxymorphone, and tapentadol, were collected. Buprenorphine [20] was excluded because it is primarily employed for treatment of opioid use disorder (OUD). Methadone when used for OUD from Narcotic Treatment Programs was also excluded. All procedures were approved by Geisinger and the University of New England’s Institutional Review Board.
- While references in the introduction used superscript numbers, references in the methods use bracketed numbers. Please standardize according to journal convention.
Response: References throughout the manuscript were standardized to brackets as indicated in the instructions for authors. A clean copy of the revised manuscript has been formatted into the MDPI standard.
- The exclusion of methadone is mentioned twice (in the first and second paragraphs) – please condense to the appropriate place.
Response: The redundant sentence was deleted. Thank you.
- The sentence “The weight in grams of each opioid 137 was obtained from ARCOS from 2010 to 2019 to isolate the peak opioid distribution year” is repeated within the second paragraph.
Response: The redundant sentence was deleted. Thank you.
- Can the authors provide more information on the generation of heat maps – which percentage change was utilized to generate these (2010 to 2019 only, or some subset in between?)
Response: The following sentence in the methods section was expanded.
Heat maps demonstrating the percent change across all 50 states per capita in total opioid distribution from 2011 to 2019 were produced using JMP V.16.2.0 software.
- Is there a reason that the authors chose the 95th:5th and 70th:30th percentile ratios? With the calculation of these ratios, I’m curious if there are other similar studies the authors can cite that used similar methodology.
Response: The results were expressed as the 95th:5th and 70th:30th percentile ratios as a measure of variation. This method of reporting variation in medication prescribing has been used by other authors. The following citation was referenced in the methods section. Further, the following sentence was added to the methods section for clarity.
The 95th : 5th percentile ratio identifies the degree of variation in distribution that each opioid experienced within a particular year across the US, where a larger ratio signifies a greater extent of variation.
- Steinman MA, Yang KY, Byron SC, Maselli JH, Gonzales R. Variation in outpatient antibiotic prescribing in the United States. The American Journal of Managed Care (2009) 15:861-868. PMID: 20001167.
- It would be great for the text to detail if the opioid distribution includes all healthcare settings, or only certain ones (does it include hospices, hospitals, etc.).
Response: The following sentence was expanded in the methods section:
ARCOS is a publicly available reporting system that reports detailed and comprehensive drug information from manufacturing to distribution (at the dispensing/retail level including hospitals, retail pharmacies, practitioners, mid-level practitioners, and teaching institutions) [17].
- For analysis of changes over time, the authors have ten years of data. What a prior analyses were planned – only looking at the differences between 2010 and 2019? Or different subsets of years in between?
Response: The following sentence was added to the methods section:
For analysis of changes over time, the a priori comparison was between the peak year for opioid distribution (2011) and 2019.
RESULTS:
- See the corresponding comment in the Table section. It would be ideal for the first paragraph of the results to link to a table with the results of the quote data per year and per drug. This appears to be the primary outcome of the manuscript, yet the reader only is given access to the % changes the authors calculated. I recognize that the authors have provided this requested information on some level in Figure 1, but given the number of opioids, these figures are not easy to read and lack the specificity that a table could provide.
Response: We apologize for the poor image quality of the figures. We have uploaded a higher resolution image for each of the figures including Figure 1. The results in the first paragraph as shown in Figure 1 should now be clearer.
- The section on ‘distribution’ is somewhat hard to follow. I’m trying to understand if there is a logical flow to how data is presented, because it is easy to get lost in a lot of random figures. Perhaps further sub-sectioning of the results would assist? Tables might also be helpful assisting so the reader has some point of reference for the text, with less numbers strewn throughout.
Response: The Distribution section of the results was reorganized and further divided into the following numbered subsections.
3.0 Results
3.1 Production
3.2. Distribution
3.2.1 National Opioid Distribution by Drug
3.2.2 State-Level, per Capita Total Opioid Distribution
3.2.3 State-Level, per Capita Variability in Opioid Distribution by Drug
3.2.4 Correlations in the per Capita MME between each Opioid
- The sentence “The 95th : 5th percentile ratio identifies the degree of variation in distribution that each opioid experienced within a particular year across 202 the US, where a larger ratio signifies a greater extent of variation” is a method and should be moved to that section.
Response: The sentence above was moved to the second paragraph in the methods section as recommended.
- As I get to the final paragraph, I’m curious whether the data needs to be sliced/diced in all the way that it has. The methods might benefit from the explanation of why all these different analyses were performed, as well as the value that each contributes to understanding the overall problem.
Response: Thank you for this comment. We hope that the reorganization of the results section with numbered section and subsection headings helps to more clearly convey the results of the study analysis.
DISCUSSION:
- The identification of distribution since 2011 – is this a novel finding? This was mentioned in the first paragraph of the introduction. I agree with the sentiment a bit further on that the particularly novel aspect of this analysis (to me) is the quota analysis, and not so much the distribution data.
Response: The discussion section was substantially reorganized with the emphasis on the analysis of production quotas in the first section.
- Ideally the discussion should not reference figures (only the methods/results).
Response: The references to figures were removed.
- Per capita analyses were done, which includes all people (regardless of whether they use opioids or not). The % of people using opioids in each state differs. I recognize this probably isn’t available in the dataset, but it might be worthwhile in the discussion to mention how this might impact interpretation.
Response: This is an important point. We added the following statement to the limitations section of the discussion.
Further, the per capita analysis was calculated using the total population and for each state rather than a subset of the population who uses prescription opioids. The proportion of persons in each state who use opioids may differ which may limit the generalizability of our analysis.
- The analysis also cannot distinguish between appropriate/inappropriate opioid use. This is a good point to mention in the limitations.
Response: This important limitation was added to the discussion as shown in the following sentence.
Importantly, this study extends upon prior ARCOS reports [4,5,9-15] and examines total opioid distribution (e.g. pharmacy, hospital, provider, and midlevel provider), although this analysis cannot distinguish between appropriate and inappropriate opioid use.
- Similar to the results, it is difficult to follow a logical flow in the discussion that help to really communicate the importance of the results. The authors do reflect and respond to a lot of their data, but the overall implications are somewhat lost as the text is quite winding in places.
Although the authors explain many of the specific trends in the discussion, an overarching ‘why is this important?’ discussion really isn’t provided. Given the importance of the opioid crisis, the government focus on supply side restrictions (without corresponding improvement in clinical outcomes), what does this study add? What value does it have? I would like to see the discussion close with some of this discussion as opposed to a simple summary of the technical findings.
Response: Thank you for this important feedback. The discussion section was substantially reorganized into the following numbered headings. The discussion for each topic was moved into the appropriate heading category to improve the clarify and flow of the discussion.
4.0 Discussion
4.1 Production
4.2 Distribution
4.3 Limitations and Conclusions
TABLES and FIGURES:
- For Table 1 of this manuscript, I’d admitted more interested in the actual amount of opioids according to year. The specific correlation factors honestly could be better suited for supplemental material, with the text highlighting important trends as the authors see fit.
It may be how the files have transferred to me as a reviewer, but the figures are very fuzzy and difficult to view. They may benefit from higher resolution as I can barely understand the information contained within. In any case, they will need to be sized much larger given the level of information that is contained within each.
I would recommend turning Figure 2A into a table to enhance readability.
Response: We apologize for the poor image quality of the figures. We have uploaded a higher resolution image for each of the figures. Please let us know if the enhanced image quality addresses the concerns above. Thank you.
Thank you for consideration of our revisions.
Respectfully,
Kenneth L. McCall, PharmD, FAPhA
Clinical Professor & Chair
Binghamton University
School of Pharmacy and Pharmaceutical Sciences
96 Corliss Ave
Johnson City, NY 13790
kmccall@binghamton.edu

Round 2
Reviewer 1 Report
Comments and Suggestions for Authors
Thank you for submitting the revised version of the manuscript. I am very satisfied. Compliments.